# OpenReview forum: "Towards Secure Tuning: Mitigating Security Risks Arising from Benign Instruction Fine-Tuning"
_ICLR.cc/2025/Conference — ICLR 2025 Conference Withdrawn Submission_

### Official Review · Reviewer_A3vn · 2024-10-28

**Soundness:** 3
**Presentation:** 2
**Contribution:** 2
**Rating:** 5
**Confidence:** 4

**Summary:**

This paper focues on preserving the safety alignment of LLMs when performing instruction tuning on benign dataset. The authors propose a solution that involves training a safety feature classifier, finding the safety-robust regions in LLMs, and fine-tuning with different learning rate. Experiments show the effectiveness of proposed method compared to vanilla instruction tuning.

**Strengths:**

- The focued phenomenon is worth-exploring.
- The idea is strightforward and ituitively feasible.
- Experiments show the effectiveness of the proposed method in preserving safety alignment compared to vanilla instruction tuning.

**Weaknesses:**

- Missing references. [1] also shows benign instruction tuning can compromise the safety alignment of LLMs. [2] also verifies safety-critical LLM parameter regions.
- Missing key baselines. [2] also searches for parameters that are critical to safety alignment, which is a key baseline for this paper. The authors should explain the advantages of their methods compared to [2] and also verify this in the experiments.
- The experiment in Table 1 is not convincing. Transforming data from AdvBench by substituting several words makes the the classifer easy to train. Then, testing on in-domain AdvBench-like data to achieve 100% accuracy is straightforward, which cannot prove too much.
- It is unclear why the authors apply such perturbations in LIne 205-209. What are the motivations behind these?

[1] Yi, Jingwei, et al. "On the vulnerability of safety alignment in open-access llms." Findings of the Association for Computational Linguistics ACL 2024. 2024.

[2] Wei, Boyi, et al. "Assessing the brittleness of safety alignment via pruning and low-rank modifications." arXiv preprint arXiv:2402.05162 (2024).

**Questions:**

no

---

### Official Review · Reviewer_RoZJ · 2024-10-30

**Soundness:** 3
**Presentation:** 3
**Contribution:** 2
**Rating:** 5
**Confidence:** 3

**Summary:**

This paper aims to mitigate the issue that large language models' (LLM) security decreases after instruction Fine-Tuning (IFT) on benign data. The authors first train a feature classifier (proxy) that discerns benign/malicious prompts at LLM's last layer. Then, they perturb different modules in the LLM to see their impact on proxy's performance on malicious prompts. The authors find some interesting patterns during the process. Based on the impact, the authors assign a large learning rate to those robust modules (named $Mods_{Robust}$) and a small learning rate to those sensitive modules during IFT on benign data. The authors name this IFT strategy as Modular Layer-wise Learning Rate (*ML-LR*) strategy. The experimental results show *ML-LR* can properly mitigate the security decrement while keeping helpfulness on two IFT scenarios (general and math-specified) over different LLMs (eg., Llama series and Vicuna series).

**Strengths:**

1. The motivation and insight of thie paper are sound. The way to detect the subset of security-robust modules is straightforward, and scanning all modules for different LLMs seems a big project.
2. The authors detect interesting patterns on security-robust modules, which can be useful for the community's future research.
3. Experimental results show *ML-LR*'s appreciate security maintance performance on benign data IFT.
4. This paper is well-structured and easy to read to me.

**Weaknesses:**

1. The proxy used to discern $Mods_{Robust}$ can be further upgraded. Sometimes the LLM can discern benign/malicious prompts, but fails to correctly react to it (e.g., refuse or not refuse)[1] (discussion on hypothesis 1 in the [1]). From this viewpoint, just discerning benign/malicious prompts without considering model's response may lead to insufficient performance when finding $Mods_{Robust}$. But it's fine to leave for future research.
2. The authors use AdvBench to train proxy, and also use it (though they may not overlap) as one of benchmarks when evaluate the IFT performance. It is better to do security evaluation on other safety-related datasets with different distributions.
3. Some figures and tables can be further polished. For example, I think the resolution of Figure 5 is a low with some words (e.g, #1->#7) seeming a little blurred; Figure. 3(b)'s layout can be further adjusted to match 3(a); And some highlights (bold or other markers) can be added on some table results (For example, Table 2. It is better to let readers to see the advantage of *ML-LR* using some markers).

[1] "On Prompt-Driven Safeguarding for Large Language Models" ICML 2024, Zheng et. al.

**Questions:**

1. In practice, more data can be used to IFT a LLM, this may lead to more training iterations. Under this case, even the learning rate of those sensitive modules is small, they may finally get updated to impact the model's security performance. Do you think this issue will take place in practice? And when you do the experiment, does this issue take place when the *ML-LR* is trained for more epcohs?
2. I notice in *Line 180*, you mention you make analaysis on 4 LLMs, and in the experiment, you use 5 LLMs (with a new Llama3.1-8b). Why analysis on Llama3.1-8b is skipped?
3. Do you ever test *ML-LR*'s performance on some dataset with a few malicious data? It might be interesting to see its performance in this case. Never mind if you have no time to do the experiment.
4. How do you think *ML-LR*'s advantage compared with those strategy that only do inference-stage intervene on LLM's modules (e.g., *RePE*[2] or training a safety suffix/prefix)?
5. See weeknesses.

[2] "Representation Engineering: A Top-down Approach to Ai Transparency" Andy Zou et. al.

---

### Official Review · Reviewer_h9Pj · 2024-11-02

**Soundness:** 2
**Presentation:** 3
**Contribution:** 3
**Rating:** 5
**Confidence:** 4

**Summary:**

The paper introduces ML-LR, a novel IFT strategy. It first categorizes different modules based on robustness and then improves model robustness by reducing the learning rates of modules more sensitive to malicious instructions. Experimental results demonstrate that ML-LR is a more robust fine-tuning method compared to IFT.

**Strengths:**

1. The proposed ML-LR method is novel.

2. The paper’s narrative is clear and logically structured, first identifying three module patterns through evaluation and then designing the IFT strategy based on these patterns.

**Weaknesses:**

1. Some results in the paper are questionable and require further explanation and correction. For example, in Table 2, why does the quality of the model’s generated responses decrease after IFT?

2. The paper only uses the LLaMA series and LLaMA-based Vicuna models. Evaluations on a broader range of models with different architectures may need to be conducted.

3. Fig. 6 shows that ML-LR is less effective than IFT against optimization-based attacks (e.g., GCG). Given that GCG is a state-of-the-art optimization-based attack, the paper should explain why this outcome occurs.

**Questions:**

1. My biggest concern is, the experimental results are not realistic. In Table 2, the quality of responses generated by BASE is even better than that of IFT and ML-LR. If that’s the case, why do we use fine-tuning? The model obtained through fine-tuning not only has reduced usability but also decreased robustness. This might imply that the quality of the fine-tuning training data is poor. The paper should at least ensure that higher quality recovery can be achieved after IFT or ML-LR.

2. The evaluation is somewhat lacking. For the general-domain scenario, the paper only tested the LLaMA series models, and for the specific-domain scenario, it only added the Vicuna series models, which are also based on LLaMA. Since the paper achieves fine-tuning by adjusting the learning rate of different parameters in the model, it should test a wider variety of LLM architectures, such as Mistral and Gemma.

3. The experimental results are also not very promising. Specifically, in Fig. 6, when the training loss of the two methods is similar, ML-LR is sometimes less effective against GCG (the state-of-the-art jailbreaking attack) compared to IFT.

---

### Official Review · Reviewer_AxNq · 2024-11-02

**Soundness:** 1
**Presentation:** 2
**Contribution:** 1
**Rating:** 3
**Confidence:** 4

**Summary:**

The authors propose fine-tuning certain components of an LLM with a lower learning rate to increase model robustness to adversarial attacks. They focus on the setting of benign instruction tuning, where the instructions themselves are not harmless.

They motivate this approach through a preliminary analysis of the effects on model safety of perturbing different model components (e.g. Q/K/V matrices) at different layers. They do this by training a harmful prompt classifier on the final embeddings of the LLM, and analyzing the drop in accuracy of this classifier when a given component is perturbed (although, as far as I can tell, they do not say how exactly they perturb the components).

The authors find that (1) components in earlier layers are more sensitive (in the sense described above) than ones in later layers, (2) Q and K matrices are more sensitive than other kinds of components, and (3) combinations of robust (i.e. non-sensitive) components can be sensitive (i.e. perturbing all components at the same time has a big impact in the classifier’s accuracy).

Based on these three findings, they propose a heuristic search routine for identifying sensitive modules inside the LLM. They then propose decreasing the learning rate of these modules by a large factor (on the order of $10^{-4}$) relative to the remaining components during fine-tuning, in effort to increase robustness to adversarial attacks. This method is referred to as Modular Layer-wise Learning Rate (ML-LR).

They then evaluate this method’s effect on general model capabilities, domain-specific capabilities, and adversarial robustness. They work with Llama 2, Llama 3 and Vicuna models, and use LoRA fine-tuning. They find that regular instruction fine-tuning (IFT) and ML-LR on general domain data produce similar results on capabilities evaluations, but ML-LR leads to lower attack success rates on many adversarial attack benchmarks. They conduct a similar study when fine-tuning on a mathematics task, and also find that the resulting capabilities are similar, but the susceptibility to attacks is lower (roughly half the attack success rate), for ML-LR compared to IFT.

**Strengths:**

1. Safety evaluations include several benchmarks (namely six), and are represented in a visually intuitive way using radar charts.

1. The authors use several open-weights models in their evaluations, indicating their results are not model-specific.

**Weaknesses:**

1. Module robustness analysis uses questionable scientific methodology: the authors pre-train a harmfulness classifier on the final embeddings of the LLM, and proceed to claim that, if a perturbation on a model component decreases the accuracy of this *same* pre-trained classifier, then this corresponds to the model safety being sensitive to this module. However, perturbing one model component can simply produce a distribution shift in the final embeddings, and need not influence top-level model behavior.

1. In fact, the three “patterns” identified by the authors on their perturbation studies could just as well be explained through the lens of distribution shift:

    - Pattern A: perturbations in earlier layers can get amplified throughout the forward pass, causing a larger distribution shift, and causing classifier accuracy to drop more. This is similar to how, in a physical system, perturbing the initial conditions can lead to larger discrepancies in the final state, as the perturbation can get amplified by system dynamics through time.

    - Pattern B: perturbations in Q and K modules impact the attention coefficients, and in particular are passed through a softmax function (as the coefficients are given by $a_{ij} = softmax(x_i^T W_Q^T W_K x_j / \sqrt{d})$), which can significantly alter the attention patterns of the model, leading to greater distribution shift.

    - Pattern C: perturbing several model components and composing them can amplify each perturbation (e.g. multiplicatively), leading to greater distribution shift.

1. Hence, I am not convinced by the author’s claim that their robustness analysis says anything about the “security” of individual model components. Crucial to my perception is the fact that they don’t retrain the harmfulness classifier after the perturbations, and simply use it as an absolute measure of model safety.

1. As highlighted above, the authors (to the best of my understanding) do not explain how they perturb components of the model in section 3.2, rendering the section unreproducible, in addition to arguably fundamentally scientifically flawed as argued above. Since this section is a key part of the paper, I consider that this constitutes enough reason for this paper to be rejected.

1. Presentation:

    - the formatting of the tables in the paper does not conform to usual standards of conference publications, e.g. using vertical bars.

    - In certain tables, e.g. Table 3, the names of models (e.g. Llama2-7B) are placed very strangely among the numerical results. In the case of Table 3, they are also incorrectly formatted (e.g. Llama213B rather than Llama2-7B or other reasonable formatting choices).

**Questions:**

1. What motivates your choice of a linear network architecture (followed by a sigmoid activation) in equation (1)? Did you observe a big difference compared to using regular logistic regression?

1. The decrease in learning rate of “sensitive components” is very significant, being as large as $10^{-4}$ in some cases. One important ablation for this method would be to set the learning rate of these modules to zero, since they are already much lower than the rest. Have the authors evaluated this alternative?

---

### Note · Authors · 2024-12-13

I have read and agree with the venue's withdrawal policy on behalf of myself and my co-authors.